# Climacteric women's perspectives on menopause and hormone therapy: Knowledge gaps, fears, and the role of healthcare advice

Marcio Alexandre H. Rodrigues[1]*, Zilma Silveira Nogueira Reis[1],
Ana Paula de Andrade Verona[2], Matheus Candido Teixeira[3], Gisele Lobo Pappa[4],
Jose Maria Soares Junior[5], Agnaldo Lopes da Silva Filho[1], Edmund Chada Baracat[5],
Gustavo Arantes Rosa Maciel[5]

**1** Obstetrics and Gynecology Department, School of Medicine, UFMG, Av. Alfredo Balena,190–2 andar, Belo Horizonte, Minas Gerais, Brazil, **2** Department of Demography, Cedeplar, UFMG, Pres. Antônio Carlos, Belo Horizonte, Minas Gerais, Brazil, **3** Federal Institute of Mato Grosso, Avenida Sen. Filinto Müller, 953, Cuiaba, Montana, Brazil, **4** Computer Science Department, UFMG, Rua Reitor Píres Albuquerque, ICEx -Belo Horizonte, Minas Gerais, Brazil, **5** Disciplina de Ginecologia, Hospital das Clinicas HCFMUSP, Faculdade de Medicina, Universidade de Sao Paulo, Sao Paulo, SP, BR. Av. Dr. Eneas, Carvalho de Aguiar, 255, andar, São Paulo, Brazil

* marcioahr@gmail.com

## Abstract

### Objective

This study aims to evaluate the knowledge, attitudes, and practices of Brazilian women regarding menopause, related symptoms, and the use of hormone therapy, including indications and contraindications.

### Methods

A cross-sectional study was conducted between 2022/07/18 and 2023/10/01, involving women aged 40–65 years from various cities in Minas Gerais, São Paulo, and other regions of Brazil. A structured KAP (Knowledge, Attitudes, and Practices) Survey was used to assess sociodemographic characteristics, the prevalence of menopausal symptoms, and the participants' knowledge and practices concerning Menopause Hormone Therapy (MHT).

### Results

The median age of the women surveyed was 50 years; 55.4% were postmenopausal (median age 55), and 44.6% were premenopausal (median age 45). The data indicate limited knowledge about menopause among Brazilian women. Less than one-third (30.3%) expressed satisfaction with the information they had received regarding menopause and available treatment options. Furthermore, 92.97% of participants demonstrated little or no knowledge of hormone therapy. Nearly 27% were unaware

**Data availability statement:** All relevant data are available from the Zenodo repository (DOI: https://doi.org/10.5281/zenodo.14867788).

**Funding:** publication fees.

**Competing interests:** M.A.H.R.- is currently receiving funding from Theramex and Besins G.A.R.M. - is currently receiving funding from Theramex J.M.S.J. - is currently receiving funding from Theramex and Pfizer The other authors have nothing to disclose.

of MHT, and 22% opposed its use. Only 29% reported current use of MHT, with common reasons for discontinuation including fear of side effects and contraindications advised by gynecologists.

## Conclusion

The findings indicate that many Brazilian women have insufficient knowledge about menopause and hormone therapy. Furthermore, a lack of information and training among healthcare providers may lead to the low utilization of menopausal hormone therapy (MHT). To effectively address menopausal symptoms and empower women to make informed choices about hormone therapy, it is crucial to improve access to accurate information and enhance the training of healthcare providers.

## Introduction

The global population is undergoing an aging phenomenon, with significant implications for various societies. According to data from the Brazilian Institute of Geography and Statistics (IBGE) in 2023, the aging process of the Brazilian population is real with around 29.29% over 50 years of the total population. [1,2]. Considering menopause usually happens at age over 45 years, we can forecast that the number of women will pass through menopausal years will increase remarkably as well as the consequence of hypoestrogenism after menopause. In fact, perimenopause women also present with symptoms that may impair the quality of life of over 40% of them [3]. Therefore, those women need treatment to relieve menopausal symptoms.

Menopause Hormone Therapy (MHT), which includes estrogens (ET), alone or associated with progestogens (EPT) is recognized as the most effective treatment of menopausal symptoms [4–6]. However, following the publication of the data from the WHI Study [7], the percentage of hormonal treatment prescriptions dropped dramatically from 85% to 18% [8]. At the onset of the 2000 decade, oral MHT usage stood at 22.4%, indicating a notable decline in prescriptions of all formulations, including isolated estrogens or those combined with progestogens. This downward trajectory persisted through 2009–2010 across all demographic groups, with current prevalence rates estimated at 4.7% overall, 2.9% for ET, and 1.5% for EPT [9].

Menopause education programs can play a vital role in increasing awareness in the world, improving attitudes, reducing impact on mental health, and enhancing the level of information and knowledge about this phase [10]. In addition to enhancing knowledge and symptom management, education programs are beneficial in addressing psychological symptoms associated, such as mood swings and anxiety [11]. Various educational approaches have been successfully employed using different modes of delivery including group and individual education sessions [10]. However, it is essential to evaluate the knowledge and perception of women in specific countries for elaborating an adequate and rational education program.

Recent studies highlight a lack of knowledge among women regarding menopause and MHT. More than half of Italian women received no information, or

described it as poor and conflicting [12], similar to Lebanese women, where 47.9% were unsure when to start therapy [13], a trend observed in other countries [14–16]. In contrast, 85.4% of Chinese women believed menopausal symptoms did not require treatment, with knowledge influenced by menopause, work, and marital status [17]. The primary sources of information for Chinese women were friends (54.7%), media (45.4%), and health professionals (2.5%) [17]. Despite prevalent symptoms, MHT use remains low. In Israel, only 29.1% of women received treatment, and 12.6% reported past or current MHT use [18]. Similar trends have been reported [12,13,17,19], with concerns over the impact of hypoestrogenism and unclear reasons for the low MHT uptake. In fact, it is relevant to evaluate the knowledge, attitudes, and myths of Brazilian women and compare them to other countries for elaborating the public politics on women's education on menopause.

Unlike two earlier studies conducted in Brazil [20,21], before the pandemic period, our survey included premenopausal and postmenopausal patients aged between 40 and 65 years. Our goal was to broaden our research to evaluate women's understanding of the physiological changes, myths, and beliefs related to hormonal shifts during the menopausal transition. We also wanted to examine how these changes influence their decision-making regarding MHT. Another significant gap is the need to update the current prevalence of MHT use among women of varying social statuses who were enrolled, particularly those who participated in face-to-face interviews within our public health system—an aspect not specifically explored in another Brazilian study [22].

While some studies have explored this topic, there are still gaps in understanding Brazilian women's perspectives on menopause, particularly concerning the issues mentioned above. This may reflect a global trend. The lack of insights hampers effective education and prevents policymakers from implementing essential initiatives to enhance the well-being of symptomatic women. We hypothesize that the menopausal transition is characterized by misinformation, fear, and deficits in understanding overall health and comorbidities, along with a lack of awareness about the benefits of hormone therapy (HT) for symptomatic women.

## Methods

### Study design, participants, and tools

This is a cross-sectional, analytical observational study using a validated tool (KAP Survey) to assess the knowledge of women residing in Brazil and selected cities worldwide (S1 Fig), regarding menopause-associated signs and symptoms, comorbidities linked to hypoestrogenism, female aging, and the use of HT. The knowledge, attitudes, and practices (KAP) methodology proposed by the World Health Organization aims to identify knowledge gaps, cultural beliefs, or behavioral patterns that can facilitate or hinder understanding and action [23]. The survey was conducted between 2022/07/18 and 2023/10/01, focusing on the general population, mainly lay women. The study was performed by two academic referral centers, Coordinating Center I, under the Gynecology and Obstetrics Department at the Medical School of UFMG, and Coordinating Center II, under the Disciplina de Ginecologia, Faculdade de Medicina, Universidade de Sao Paulo, Sao Paulo, SP, BR.

Data collection occurred through online surveys using a specific questionnaire issued by both centers and face-to-face interviews. Women were eligible for inclusion if they were between 40 and 65 years old. Exclusion criteria were women currently undergoing chemotherapy or radiotherapy for cancer treatment and those with cognitive or visual impairments that prevented them from completing the questionnaire. We also conducted face-to-face interviews, using the same online questionnaire, with women attending routine consultations in our public health service. This approach aimed to accurately represent a broader range of lower-income patients, including those who are illiterate. Fig 1 presents a flowchart illustrating the sampling process.

This approach, which consists of a set of issues adaptable to various scenarios of interest, supports the identification of barriers that oppose efforts to control a particular public health problem. The conceptual basis of the approach, applied to the context of this study, has the following structure: Knowledge: what a woman already knows about menopause, and

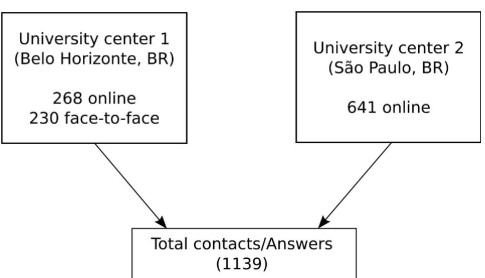

**Fig 1. Flow diagram reporting sampling process.**

MHT; Attitude: what the woman believes is best for her health and how she intends to position herself concerning menopause and therapy opportunities; Practice: what symptoms bothers her and what treatments she has used in the past or currently uses.

We conducted a pilot study of the KAP survey to evaluate the clarity and comprehensibility of the proposed questions and answers. This involved a group of three physicians specializing in menopausal care and a sample of ten women from the same age group as the target demographic. The responses from these women were excluded from the final analysis.

The interviews and data collection characterizing the sample of women were collected with the support of software, using the following strategies: direct collection in digital medium supported by a mobile device with internet access (Google Forms ™); collection in paper form transcribed by the researcher for the virtual environment, in case the internet access was not possible.

Each woman was asked about sociodemographic data and some aspects of their health status. We used the following definition to characterize the menopausal status: natural menopause - permanent cessation of menstrual periods, diagnosed after 12 months of amenorrhea, after 40 years of age; surgical menopause - menopause caused by the removal of both ovaries; perimenopause - period preceding menopause characterized by variations in the menstrual cycle [24].

The survey instrument consisted of 3 parts with 55 questions, i.e., knowledge about menopause-related symptoms and comorbidities; attitudes, including issues such as whether they would take MHT or not to treat menopausal symptoms and practice aspects such as menopause-symptoms related intensity, if they are currently using or had ever taken MHT, and for how long time. We used a Likert scale in some questions where we aimed to assess, for example, the degree of satisfaction with the information received of the level of Knowledge on various aspects of menopause and its treatment. For more details, refer to the supplementary material, tables, and the Portuguese version of the questionnaire [S4 Table].

**Sample size calculation.** It was based on the primary outcome. In the study by Pacello et al., involving Brazilian women, an expected misinformation rate about MHT of 60.2% was observed [20]. Assuming an infinite population size of women within the target age group in Brazil, and a 3.5% difference between the expected proportion of 60.2% and the observed proportion, with a 95% confidence level, the sample size was estimated to be 751 women. To account for an anticipated dropout rate of 15% post-recruitment, at least 864 women ~~will~~ would be required to conduct the study. The sample size calculation software used was EPFI Info from the CDC in Atlanta [Epi Info, version 7.2.3.1], utilizing the StatCalc function for Population Surveys. Sampling was obtained through convenience, simultaneously at the two participating centers. This non-probability sampling method was chosen due to its practicality and efficiency in reaching a significant number of participants within the desired demographic and geographic constraints. The sampling technique was deemed appropriate for this exploratory phase of the research, aiming to approach the KAP of Brazilian women regarding menopause, symptoms related to, and further, HT use.

**Data analysis.** The characterization of the general study group was described according to the nature of the variable. Categorical variables were presented in absolute and relative proportions and quantitative variables, were presented as measures of central tendency and variability. The normality of quantitative variables was evaluated by applying the Kolmogorov-Smirnov test. The association between variables was verified by the chi-square or Fisher's exact tests (when indicated) for categorical variables, the Student's t-test for quantitative variables with normal distribution, and the Mann-Whitney test for quantitative variables without normal distribution. The significance level for hypothesis tests is $p < 0.05$. The SPSS software and Phyton LibrarySciPy were for data analysis. For the univariate analysis of qualitative variables, the study used frequencies, percentages, and measures of central tendency and dispersion for quantitative variables. The study used a $p$-value for the bivariate analysis, considering a value of $p < 0.05$, OR, confidence intervals, and Pearson's chi-square. The influence of all variables simultaneously was evaluated with a logistic regression model. Further information is described in the Supplemental Material Section.

**Ethical considerations.** The project has been approved by the Ethics Committee of Hospital das Clinicas HCFMUSP, Faculdade de Medicina, Universidade de Sao Paulo, Sao Paulo, SP, BR., # CAAE 55644522.2.0000.0068, # Consolidated opinion by the REC (Research Ethics Committee): 5,234,965 and complies with the Declaration of Helsinki. All patients provided written informed consent before answering the questions.

## Results

All 1,139 women interviewed for the study, were included in the final analysis according to the University center (Fig 1). They were sorted into two groups: premenopausal (menopausal transition) and postmenopausal. 31.6% were interviewed face-to-face design only in University Center 1, while the remaining participants completed an online Google- forms™. A geography overview of the residence cities of women participating in the study is presented on a map as S1 Fig.

The baseline characteristics of women enrolled are displayed in Table 1. The median age of all women surveyed was 50 years, 55.4% were postmenopausal (median age of 55), and 44.6% were premenopausal (median age of 45). There was no statistical difference in baseline characteristics between, ethnicity, and educational degree, between both groups.

The p-values presented are from the chi-square statistical test. The bold lines indicate a question in the question-naire and the lines that follow it are the corresponding alternatives. The N column indicates the number of women who answered the question. The "Total" column indicates the number of women who selected a given alternative. The "Pre/Post-menopausal women" columns indicate the number of women who marked a given option who are pre-menopausal or post-menopausal. The absolute number of responses for each question does not represent the total sample of 1,139 participants due to missing data.

The characteristics of women according to menopausal status are displayed in Table 2. Most comorbidities self-reported by women surveyed were significantly more prevalent in post-menopausal women than in premenopausal, namely hypertension (46.1%), diabetes (19.9%), and cardiovascular disease (14.4%) also seen in S2 Fig

## Knowledge

Data on menopause knowledge, categorized by menopause status, are shown in S1 Table. Women in the post-menopausal group (98.25%) were significantly more knowledgeable about symptoms than pre-menopausal women (89.29%) ($p < 0.001$). Hot flashes were the most recognized symptom, cited by nearly all respondents. Other symptoms, such as vaginal dryness, mood swings, sleeplessness, and loss of libido, were better recognized by post-menopausal women ($p < 0.001$) (S1 Table). Additionally, menopausal women more frequently cited doctors as their main information source (82.3%) compared to pre-menopausal women (59.7%), who mainly relied on relatives or friends (76.9%) ($p < 0.001$) (Fig 2A1 and 2A2).

2A: How did you find out about menopause symptoms? 2A1 Menopause group - 2A2 Premenopause group. 2B: How satisfied were you with the information on menopause that your physician had given? 2B1 Menopause group – 2B2 Pre-menopause group

**Table 1. Baseline characteristics of the women by menopausal status.**

| Question | N | Total (%) | Post-menopausal women | Pre-menopausal women | p-value |
|---|---|---|---|---|---|
| **Ethnicity (self-reported)** | 1137 | | | | **0.070** |
| White, n/N (%) | | 545 (47.93%) | 330 (52.38%) | 176 (44.00%) | |
| Brown, n/N (%) | | 452 (39.75%) | 231 (36.67%) | 170 (42.50%) | |
| Black, n/N (%) | | 110 (9.67%) | 54 (8.57%) | 41 (10.25%) | |
| Yellow, N (%) | | 26 (2.29%) | 14 (2.22%) | 10 (2.50%) | |
| Indigenous, n/N (%) | | 4 (0.35%) | 1 (0.16%) | 3 (0.75%) | |
| **Marital status** | 1139 | | | | **0.020** |
| Married, N (%) | | 637 (55.93%) | 365 (57.84%) | 211 (52.62%) | |
| Single N (%) | | 63 (5.53%) | 30 (4.75%) | 27 (6.73%) | |
| Divorced, N (%) | | 267 (23.44%) | 155(24.56%) | 87 (21.70%) | |
| Cohabitation/consensual marriage/common-law marriage, N (%) | | 172 (15.10%) | 81 (12.84%) | 76 (18.95%) | |
| **Educational degree** | 1139 | | | | **0.452** |
| Elementary school, nN (%) | | 69 (6.06%) | 46 (7.29%) | 21 (5.24%) | |
| Incomplete elementary school, N (%) | | 104 (9.13%) | 61 (9.67%) | 34 (8.48%) | |
| High school, N (%) | | 327 (28.71%) | 174 (27.58%) | 115 (28.68%) | |
| Incomplete high school, N (%) | | 92 (8.08%) | 54 (8.56%) | 31 (7.73%) | |
| Graduation, N (%) | | 210 (18.44%) | 113 (17.91%) | 81 (20.20%) | |
| Incomplete graduation, N (%) | | 45 (3.95%) | 26 (4.12%) | 11 (2.74%) | |
| Post-graduation, N (%) | | 288 (25.29%) | 154 (24.41%) | 108 (26.93%) | |
| Not specified, N (%) | | 4 (0.35%) | 3 (0.48%) | 0 (0%) | |
| **Occupation** | 1135 | | | | **< 0.001** |
| Formally employed, N (%) | | 485 (42.73%) | 235 (37.24%) | 207 (52.14%) | |
| Informally employed, N (%) | | 306 (26.96%) | 161 (25.52%) | 109 (27.46%) | |
| Unemployed, N (%) | | 186 (16.39%) | 102 (16.16%) | 61 (15.36%) | |
| Retiree | | 158 (13.92%) | 133 (21.08%) | 20 (5.04%) | |
| **Health Insurance** | 1139 | | | | **0.6143** |
| Public | | 498 (43.72%) | 270 (42.79%) | 167 (41.65%) | |
| Private | | 423.0 (37.14%) | 233 (36.93%) | 159 (39.65%) | |
| Private and public | | 217 (19.05%) | 128 (20.29%) | 74 (18.45%) | |
| None of them | | 1 (0.09%) | 1 (0.25%) | 0 (0%) | |

Women were asked how satisfied they were with the information on menopause that their physician had given. Less than one-third (30.3%) of all women surveyed reported they were "*completely satisfied*" with the information they had received. Menopausal women were more satisfied taking together those who responded "Completely *satisfied*" or "*Reasonably satisfied*" than the pre-menopausal ones (64.7% and 48.0%, respectively) these differences were statistically significant ($p < 0.001$) (Fig 2B1 and Fig 2B2).

Comparing knowledge about a list of diseases related to menopause, osteoporosis is more known than cardiovascular disease in both groups, and all diseases, except osteoporosis, were more known by postmenopausal women, than pre-menopausal ones, which was statistically significant ($p < 0.001$) (S1 Table).

**Table 2. General characteristics of women according to menopausal status (n = 1139).**

| Characteristics | Total (n = 1139) | Postmenopausal women (n = 631, 55.40%) | Pre-menopausal women (n = 508, 44.60%) | p-value |
|---|---|---|---|---|
| Had Children, n (%) | 954 (83.76) | 527 (83.52) | 427 (84.06) | 0.807# |
| Number of children, median (IQR) | 2 (2.00) | 2 (2.00) | 2 (1.00) | 0.503* |
| Age of the first childbirth, median (IQR) | 24 (10.00) | 25 (10.00) | 23 (12.00) | <0.001* |
| **Comorbidities (self-reported)** | | | | |
| Hypertension, n (%) | 433 (38.02) | 291 (46.12) | 142 (27.95) | <0.001# |
| Diabetes, n (%) | 199 (17.47) | 126 (19.97) | 73 (14.37) | 0.013# |
| Cardiovascular diseases, n (%) | 119 (10.45) | 91 (14.42) | 28 (5.51) | <0.001# |
| Osteoporosis, n (%) | 69 (6.06) | 62 (9.83) | 7 (1.38) | <0.001# |
| Thrombosis, n (%) | 35 (3.07) | 22 (3.49) | 13 (2.56) | 0.368# |
| Stroke, n (%) | 19 (1.67) | 15 (2.38) | 4 (0.79) | 0.037# |
| Cancer, n (%) | 39 (3.42) | 27 (4.28) | 12 (2.36) | 0.077# |
| I don't know, n (%) | 572 (50.22) | 259 (41.05) | 313 (61.61) | <0.001# |
| **Gynecological Care (last visit)** | | | | **0.985#** |
| Less than 1 year, n (%) | 650 (57.07) | 375 (59.43) | 275 (54.13) | |
| Between 1–3 years, n (%) | 339 (29.76) | 175 (27.73) | 164 (32.28) | |
| Between 3–5 years, n (%) | 69 (6.06) | 31 (4.91) | 38 (7.48) | |
| More than 5 years, n (%) | 36 (3.16) | 21 (3.33) | 15 (2.95) | |
| More than 10 years, n (%) | 42 (3.69) | 28 (4.44) | 14 (2.76) | |
| Never, n (%) | 3 (0.26) | 1 (0.16) | 2 (0.39) | |

*Kruskal-Wallis Test.

#Chi-square Test.

Analyzing components of the KAP survey model, we present the outcomes in three subsections.

Women were asked regarding their level of concern with complaints related to the menopausal period and some comorbidities. The risk of breast cancer was most frequently mentioned by approximately 49% of women as one greatest concerns to them (including *"I´m worried a lot" and "I'm extremely worried"*). Further, emotional complaints, cardiovascular disease, and osteoporosis were reported, respectively, as the second, third, and fourth disorders that concern all women (Fig 3). In Fig 3, the total number of responses is 653. However, in (S1 Table), which displays the varying levels of concerns regarding complaints related to menopause and certain comorbidities among postmenopausal and premenopausal women, the number of responses is 1,133. The difference in these numbers arises because, in the S1 Table, all participants responded to the question, while in Fig 3, the data is based on responses to conditional questions from 19 to 26 (S4 Table).

Comparing the two groups, postmenopausal women had more knowledge, with statistical significance (p < 0,001), about the hormonal "*status*" of the menopausal period than premenopausal, and unexpectedly 26.3% don´t know which hormones the ovaries produce. ~~Not surprisingly~~, Around 16% of premenopausal women don't know what happens with the hormones after menopause, which was statistically significant when compared with the postmenopausal women (p < 0,001) (S1 Table).

Asking women if there is a treatment for menopausal symptoms, the majority of those currently going through menopause (91,9%) answered yes. Further, more than 80% of women surveyed cited HT as one of the treatments for menopausal symptoms, and less than 3% answered that didn´t know (S1 Table).

Querying what women know about MHT, ~~remarkably~~ 92.97% of them had little or no knowledge. Premenopausal women were more likely than postmenopausal to have little or no information (Fig 4). Despite scanty knowledge about

Figure 2A1 : Post-menopausal women

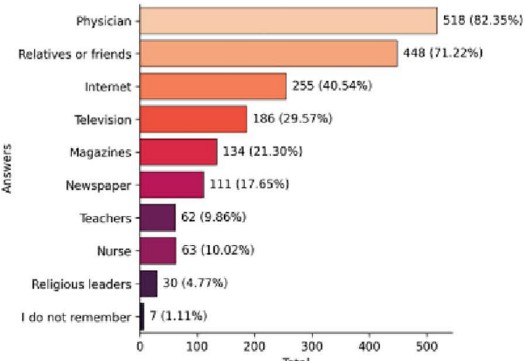

Figure 2A2 : Pre-menopausal women

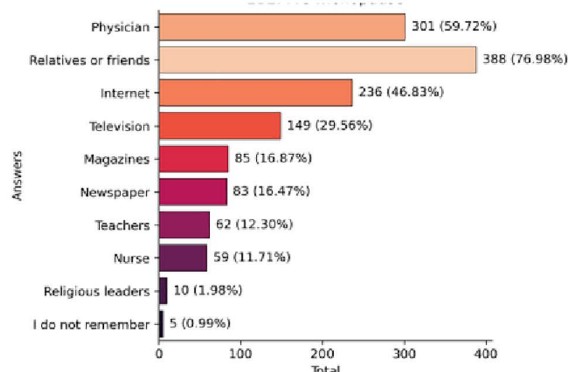

Figure 2B1: Post-menopausal women

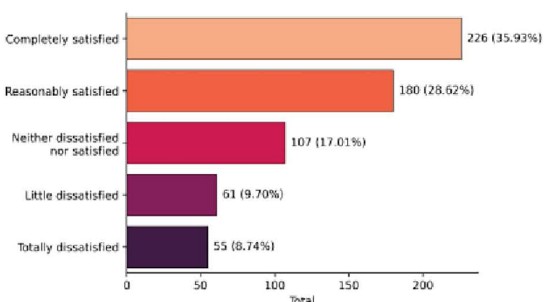

Figure 2B2: Pre-menopausal women

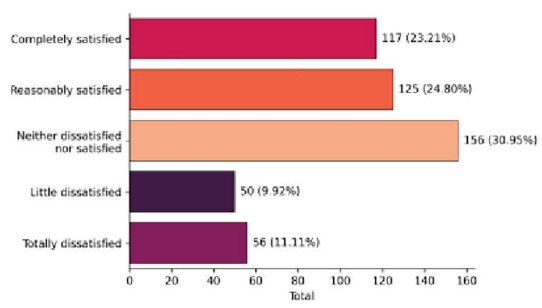

**Fig 2. Knowledge of menopause, according to menopause conditions.**

it, most of them (62.7%) answered that there are negative effects of using hormones, which in a way makes evident a discrepancy in these answers. Further, postmenopausal women (71.6%) were more likely to cite this information than premenopausal (51.7%) (S1 Table).

## Attitudes

The attitudes of women relating to MHT are displayed in S2 Table. Most postmenopausal women (25.51%) were less prone to use MHT than premenopausal (15%), which was statistically significant (for those that responded *"wouldn´t take or don´t think they would take MHT"*) ($p < 0.001$) (Fig 5). Further, most postmenopausal women (38%) responded with a higher probability of using HT to treat symptoms (*"I´m sure, I would take"*) than premenopausal (25.9%), with statistical significance ($p < 0,001$).

The main reasons for postmenopausal women not to use HT to treat symptoms were fear of side effects (65.15%) and unfavorable physician opinion (13.28%). For pre-menopausal women, the most common reason was also potential adverse events (58.55%). Doctors' opinions were less important (5.92%) to this group than to postmenopausal ones (13.28%), which was statistically significant ($p < 0,001$) (Fig 6A and 6B). It is important to mention that almost one-third of premenopausal women reported that they do not know why they would not take MHT to treat menopausal symptoms.

Of those women with indications to use MHT, around 61% of them received this recommendation from a gynecologist. However, fewer premenopausal women (33.3%) had received this recommendation than postmenopausal ones (69.9%),

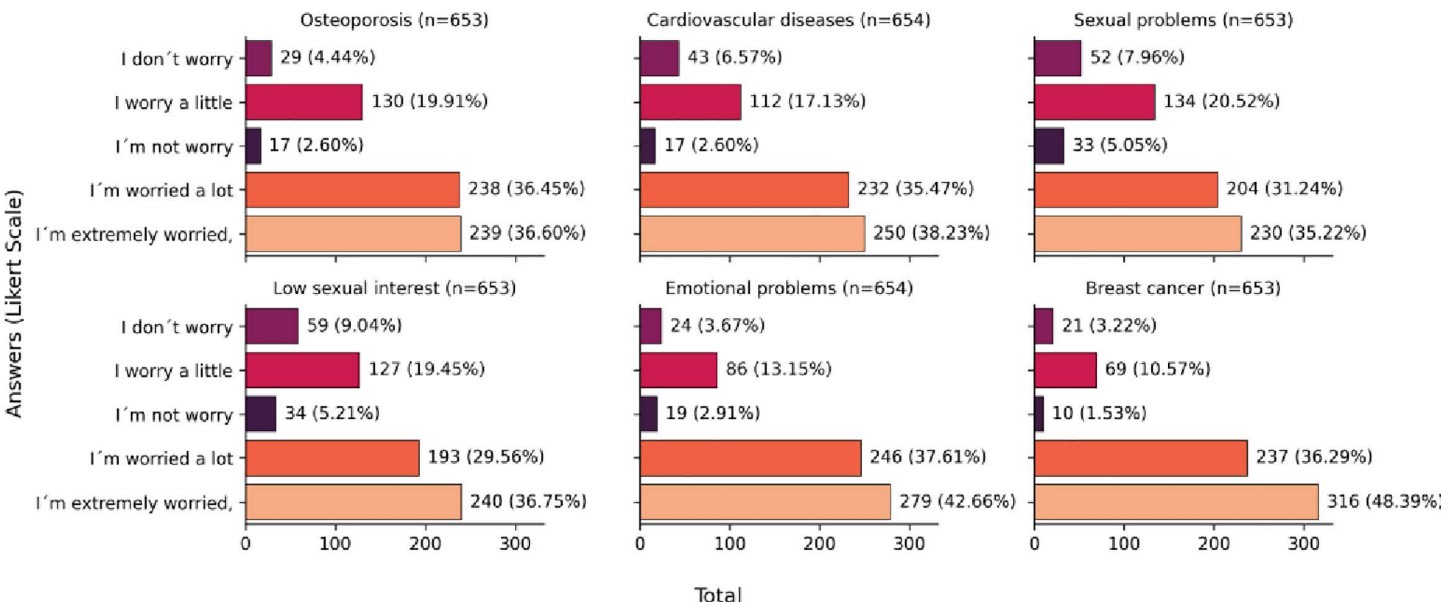

**Fig 3. Level of concern with complaints related to the menopausal period and some self-rated comorbidities (n = 1133).**

Figure 4A: Menopause

Figure 4B: Pre menopause

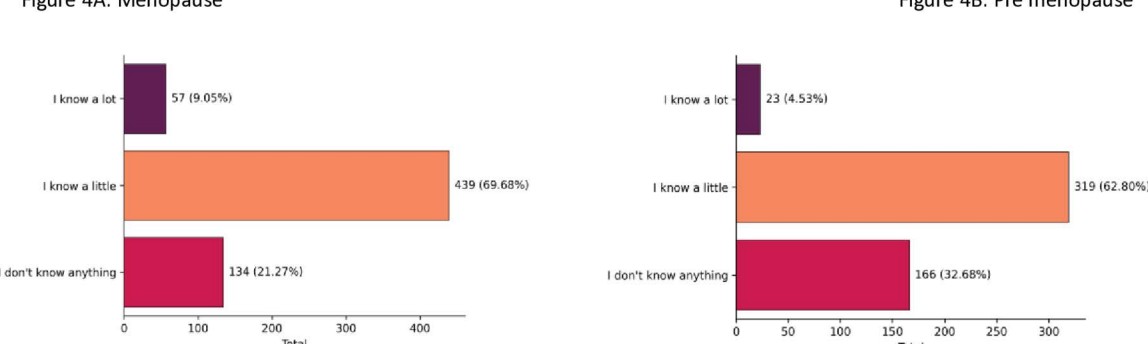

**Fig 4. Knowledge of menopause, according to menopause conditions.** What do you know about Menopause Hormone Therapy (MHT).

which was statistically significant ($p<0.001$). HT, physical activities, and herbal medicines were the most frequent treatment options discussed by gynecologists reported by both groups (S2 Table).

The most frequently chosen reasons or indications to initiate HT by postmenopausal women were to relieve vasomotor symptoms (74%), to improve quality of life (61.4%), and to relieve urogenital symptoms (54.5%). Around 38% of them mentioned osteoporosis and 29% cardiovascular disease prevention as other reasons. The statistics for premenopausal women were quite similar (S2 Table and Fig 6C and D).

## Practices

Practice aspects were assessed from all 631 postmenopausal women (S3 Table). In 85.9% of them, menopause occurs spontaneously. The three most quoted symptoms cited by menopausal women were vaginal dryness, sleeplessness, and

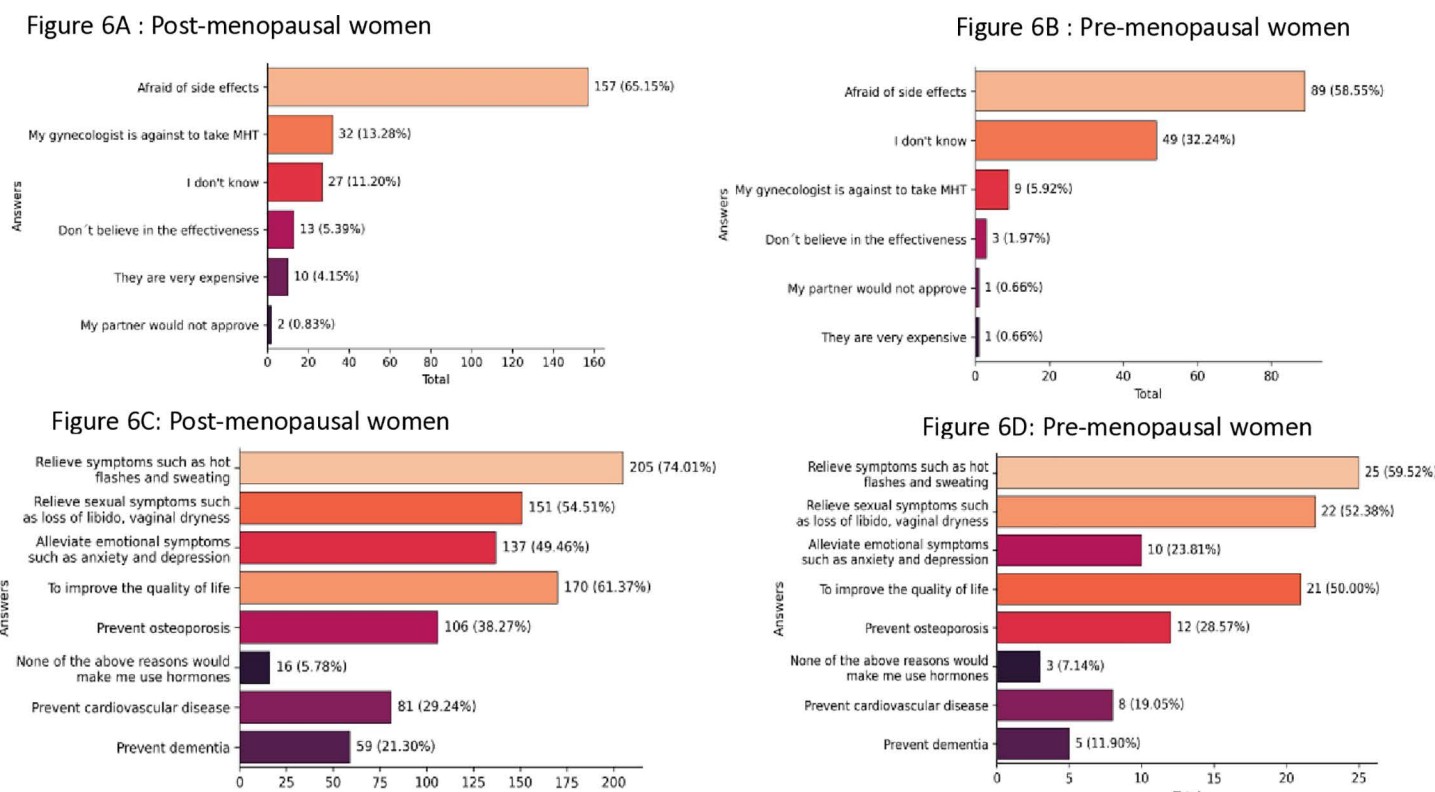

**Fig 5. Reasons mentioned by women would not use (MHT: menopause hormone therapy).**

**Fig 6. Attitudes of women regarding MHT (MHT: menopause hormone therapy).** 6A and 6B. Reasons for women not to use HT to treat symptoms. 6C and 6D. Reasons or indications mentioned by women to initiate MHT.

urine loss during physical activity. However, vasomotor symptoms (VMS) were in sixth place among symptoms presented in a previous list (Fig 7).

Regarding VMS, 55% related high frequency ("*very often*" or "*always*") of which approximately 23% claimed this burden symptom as every time in their lives. Concerning sexual aspects and urinary symptoms, 68.46% reported vaginal dryness, 48.65% pain during intercourse, 39.9% burning sensation in the vagina, and 51.35% loss of urine during physical activities. Further, 65.45% of them complained of sleep disrupted by urinary urgency (S3 Table).

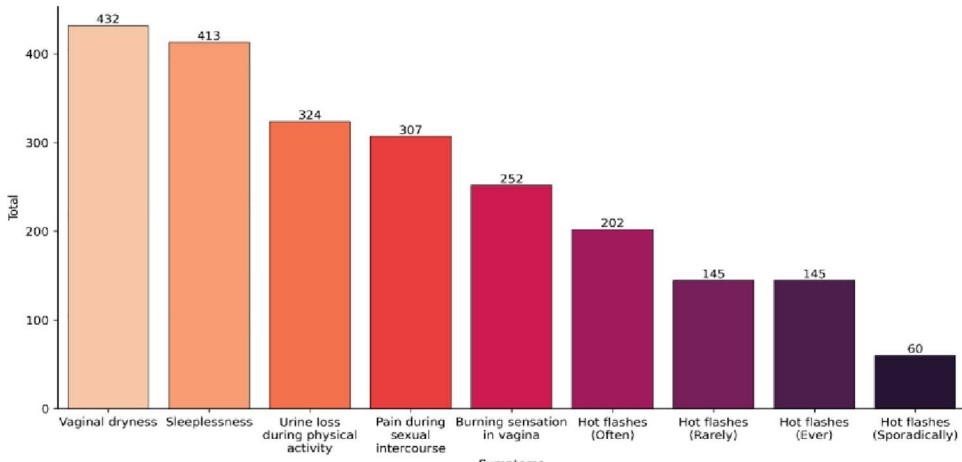

**Fig 7. Symptoms referred by women enrolled in The Survey (n: 631 out of 1139 interviewed) (also in S3Table).**

Only 29.32% informed that they were currently taking MHT, and 15,37% were past users. Around 55% of postmeno-pausal symptomatic women never used HT. Concerning how long they had been taking hormones to treat symptoms, 27.95% informed that they had used it for less than 1 year, and 58.39% between 1 and 5 years. Specifically regarding reasons women stop using hormones, the most common was the doctor's decision, 15.96%, and the second reason (11,7%) was side effects. Only 3.5% stopped for fear of cancer (S3 Table).

Our results highlight an important observation regarding women's usage of hormone therapy (HT) across different healthcare systems. Specifically, 62.59% of women who rely on the public health system (from a sample of 270) are less likely to use HT, compared to 49.78% of those in the private system (from a sample of 233) representing a difference of 12.81%. To assess the statistical significance of this observation, we performed a Chi-Square test, which yielded a p-value of 0.0168. Given a significance level of 0.05 (95% confidence), we can reject the null hypothesis and conclude that these results are unlikely to be due to chance. Further research will explore the broader impact of public and private health systems on treatment choices.

## Discussion

A KAP survey is widely used in public health to assess gaps between knowledge and actual practices, crucial for understanding treatment adherence [25]. Our study is the first Brazilian analysis using the KAP survey tool on women's knowledge, attitudes, and practices towards menopause and HT. All data suggested that the levels of satisfaction and knowledge, especially regarding HT may be low. This fact may impact treatment adherence [25]. The WHO validates the KAP tool [21], and our findings offer key insights into how Brazilian women approach menopause and HT adherence.

### Knowledge

Our data demonstrated limited knowledge on the menopause among Brazilian women. Nearly 27% were unaware of HT, and 22% opposed its use. Also, only 29% reported current HT use, with common reasons for discontinuation being fear of side effects and contraindications by gynecologists. This highlights the role of knowledge in influencing patient attitudes and practices. In addition, the post-menopausal women were better informed about symptoms (98.25%) than pre-menopausal women (89.29%), a significant difference, possibly due to their lived experiences. The results contrast with a UK study, where post-menopausal women reported higher levels of knowledge [26], while Chinese women seemed

to have an even higher awareness [17]. Our findings are in accordance with another Brazilian study [22] but also indicate persistent gaps in knowledge, especially when compared to international data. In clinical practice, the low knowledge affects the reason for using the hormonal therapy as well as the adherence. The other point is the fear of misunderstanding or lack of clarity about what is really happening with women during the climacteric phase.

VMS and sleep disturbances were the most commonly cited menopausal symptoms, consistent with other studies [19,22]. Women listed osteoporosis and cardiovascular disease (CVD) as their top health concerns, although global morbidity data suggest CVD should be prioritized [27]. Misconceptions about CVD prevalence, likely due to a lack of awareness campaigns, may explain this discrepancy. This fact is a concern for the health of women during this period.

Medical Doctors were cited as the primary source of information by 70% of respondents, aligning with findings from Brazilian [22] and Chinese studies [17]. In the UK [26], fewer women are turning to doctors for information, preferring to use websites instead. This trend raises concerns. In Brazil, physicians could play a crucial role in providing information and should receive better training to assist menopausal women. Additionally, the impact of social media is growing worldwide. It may be necessary for physicians to adapt to this new landscape, helping women enhance their understanding and management of menopause knowledge.

Regarding HT, 69% of menopausal and 62.8% of pre-menopausal women reported some knowledge. This mirrors another Brazilian study [22]. Interestingly, only 3.5% cited fear of cancer as a reason to discontinue HT, diverging from previous studies that emphasize cancer risk [28]. The concern of the HT impact on the health may influence the decision of physicians. In fact, the gynecologist's recommendation was a significant factor in the discontinuation of HT, which is a novel finding from our Survey. One possible explanation for this could be the fear that patients may develop health issues like thrombosis or cancer as a result of hormone therapy prescriptions. Physical activity was another common treatment known by 88.26% of menopausal women. This aligns with a UK study, where exercise was a prevalent non-pharmaceutical option [26]. However, HT remains the most recognized treatment among Brazilian women.

## Attitudes

Women showed more positive attitudes toward menopause and its treatments compared to their knowledge about the subject. Most participants discussed the importance of HT and highlighted physical activity as a key factor for their health. This finding is consistent with an American survey in which 84% of respondents reported discussing HT during medical evaluations [19]. However, other data from an Italian study revealed a different perspective: 53% of women reported receiving more information about menopause than about HT (39%), and among those who were informed about HT, 57% received conflicting information [12]. Despite HT being the most effective treatment for VMS, misconceptions persist, leading to unmet demands. Our survey found that post-menopausal women were more likely to use HT (25%) than pre-menopausal ones (15%), potentially because those experiencing symptoms seek relief more actively. Other limiting factors may include the length of consultations, lack of specialized menopause care, and unavailability of hormone drugs in public health services in our country.

When asked about their reasons for starting HT, 72% of respondents cited symptom relief, followed by improvements in quality of life (59%) and sexual concerns (54%). These findings are consistent with existing literature [17]. Prevention of osteoporosis was mentioned by 37% of participants, while 28% cited prevention of cardiovascular disease; these figures are also similar to those found in another study [19]. Among our post-menopausal women surveyed, the primary reasons for avoiding HT included concerns about adverse events (65%) and opposition from gynecologists (13%). Following the Women's Health Initiative (WHI) study [7], many women worldwide chose to stop HT; for example, in the U.S., 93% of patients were less likely to request it [29–31]. Even two decades after the WHI results and numerous re-evaluations of the data—taking into account new concepts like the "window of opportunity" for initiating HT—our findings indicate that Brazilian gynecologists still exhibit a degree of resistance to prescribing it. This hesitance affects their decisions regarding the treatment of symptomatic women. The data presented may serve as a call to action to change the current approach to HT prescription and better serve women's health needs.

## Practices

Of 631 postmenopausal women, the most reported symptoms were vaginal dryness, sleep disturbances, and urinary leakage during physical activity. Notably, 23% of women experienced frequent daily vasomotor symptoms. This aligns with a meta-analysis by Fang et al. [32], which identified joint and muscular discomfort (65.43%), physical and mental exhaustion (64.13%), and hot flashes (52.65%) as the most common complaints globally. Urogenital symptoms were also prevalent, with sexual problems (53.97%), vaginal dryness (44.81%), and urinary issues (40.27%) being the most reported. Although our study did not include somatic symptoms, the high prevalence of urogenital complaints is consistent with findings from the VIVA-LATAM Survey [33] and contrasts with studies focusing on VMS [12].

The prevalence of urogenital complaints was notably high: 68% reported vaginal dryness, 48% experienced pain during intercourse, nearly 40% had a burning sensation, and 51% had urine leakage during sexual activity. This underlines the often overlooked nature of these symptoms in routine consultations [34,35]. Our findings suggest that urogenital symptoms are significantly bothersome and frequently underreported, supporting the need for better screening and management of these issues.

Regarding HT, 55% of symptomatic postmenopausal women never used it, 29% were current users, and 15% had used it previously. This is comparable to American data [19] but higher than Italian findings [12] and much higher than in China [17]. Despite a high frequency of moderate to severe symptoms, about a third of women had not received treatment [36]. Our data indicate a lower prevalence of HT use compared to previous studies [20,37]. This suggests that economic factors and comorbidities may influence access to and utilization of HT, in addition to the fear of cancer and the stance of gynecologists against HT.

Our study has several strengths, including a large sample size of 1,139 women, which includes illiterate and low-income populations. We utilized a validated tool from the World Health Organization (WHO) and allowed participants to respond through multiple formats, including online surveys and face-to-face interviews. However, there are also some limitations. Our sample was predominantly composed of individuals from higher socioeconomic backgrounds, and our reliance on digital questionnaires may not have reached all demographic groups.

## Conclusion

Our data indicate that it is essential to provide quality education to improve understanding of menopause. Enhancing communication between healthcare providers and women can increase knowledge and help dispel misinformation surrounding menopause and hormone therapy. It is crucial to ensure that quality, accessible information is available, along with effective training for healthcare professionals, to manage menopausal symptoms and hormone therapy effectively. Additionally, further research is needed to explore how social media can influence the knowledge, attitudes, and practices of women in the post-menopause period.

## Supporting information

**S1 Fig. Overview of residence cities of women participating in the study.**
(TIF)

**S2 Fig. Comorbidities self-rated by women surveyed according to menopausal status.**
(TIF)

**S1 Table. Knowledge of menopause, according to menopause condition.**
(DOCX)

**S2 Table. Attitude of women, according to menopause mode of assessment.**
(DOCX)

**S3 Table. Practices of women in post-menopausal women (n = 631).**
(DOCX)

**S4 Table. Survey with questions in Portuguese version.**
(DOCX)

## Author contributions

**Conceptualization:** Marcio Alexandre H. Rodrigues, Zilma Silveira Nogueira Reis, Ana Paula de Andrade Verona, Gustavo Arantes Rosa Maciel.

**Data curation:** Marcio Alexandre H. Rodrigues, Matheus Candido Teixeira, Gisele Lobo Pappa, Gustavo Arantes Rosa Maciel.

**Formal analysis:** Marcio Alexandre H. Rodrigues, Matheus Candido Teixeira, Gisele Lobo Pappa.

**Funding acquisition:** Marcio Alexandre H. Rodrigues, Gustavo Arantes Rosa Maciel.

**Investigation:** Marcio Alexandre H. Rodrigues, Gustavo Arantes Rosa Maciel.

**Methodology:** Marcio Alexandre H. Rodrigues, Zilma Silveira Nogueira Reis, Ana Paula de Andrade Verona.

**Project administration:** Marcio Alexandre H. Rodrigues, Ana Paula de Andrade Verona.

**Resources:** José Maria Soares Junior, Agnaldo Lopes da Silva Filho, Edmund Chada Baracat.

**Supervision:** Marcio Alexandre H. Rodrigues.

**Validation:** Marcio Alexandre H. Rodrigues.

**Writing – original draft:** Marcio Alexandre H. Rodrigues, Zilma Silveira Nogueira Reis, Gustavo Arantes Rosa Maciel.

**Writing – review & editing:** Marcio Alexandre H. Rodrigues, Zilma Silveira Nogueira Reis, Ana Paula de Andrade Verona, José Maria Soares Junior, Agnaldo Lopes da Silva Filho, Edmund Chada Baracat, Gustavo Arantes Rosa Maciel.

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
