## [Decision Letter · Decision Letter 0]

10 Jan 2025

PONE-D-24-57399Climacteric Women’s Perspectives on Menopause and Hormone Therapy: Knowledge Gaps, Fears, and the Role of Healthcare Advice.PLOS ONE

Dear Dr. Rodrigues,

Thank you for submitting your manuscript to PLOS ONE. After careful consideration, we feel that it has merit but does not fully meet PLOS ONE’s publication criteria as it currently stands. Therefore, we invite you to submit a revised version of the manuscript that addresses the points raised during the review process.

**ACADEMIC EDITOR: ** Congratulations on the study. The work has the potential to advance knowledge about the epidemiology of climacteric syndrome among Brazilian women. Overall, the reviewers' comments were positive, with suggestions for minor revisions throughout the text.

As an academic editor, I believe it would be very interesting if the authors could stratify the analyses not only by menopausal status, but also by the type of health system use, i.e., public vs. private. Approximately 50% of the sample is composed of users of the public system and 50% of users of the private system. It would be important, for example, to know the prevalence of MHT use among users of the Brazilian public system and compare it with the private system. In addition, to know the opinions about MHT comparing these two groups of women. It was not clear to this reader the difference between “Private w/ Healthcare” and “Private w/o Healthcare”.

We look forward to receiving your revised manuscript.

Kind regards,

Luiz F. Baccaro

Academic Editor

PLOS ONE

Journal Requirements:

M.A.H.R.- is currently receiving funding from Theramex and Besins

G.A.R.M. - is currently receiving funding from Theramex

J.M.S.J. - is currently receiving funding from Theramex and Pfizer

The other authors have nothing to disclose. 

5. We note that Figure S1 in your submission contain [map/satellite] images which may be copyrighted. All PLOS content is published under the Creative Commons Attribution License (CC BY 4.0), which means that the manuscript, images, and Supporting Information files will be freely available online, and any third party is permitted to access, download, copy, distribute, and use these materials in any way, even commercially, with proper attribution. For these reasons, we cannot publish previously copyrighted maps or satellite images created using proprietary data, such as Google software (Google Maps, Street View, and Earth). For more information, see our copyright guidelines: http://journals.plos.org/plosone/s/licenses-and-copyright.

a. You may seek permission from the original copyright holder of Figure S1 to publish the content specifically under the CC BY 4.0 license.  

Reviewers' comments:

Reviewer's Responses to Questions

**Comments to the Author**

1. Is the manuscript technically sound, and do the data support the conclusions?

Reviewer #1: Yes

Reviewer #2: Yes

Reviewer #3: Yes

2. Has the statistical analysis been performed appropriately and rigorously? 

Reviewer #1: Yes

Reviewer #2: Yes

Reviewer #3: Yes

3. Have the authors made all data underlying the findings in their manuscript fully available?

Reviewer #1: Yes

Reviewer #2: Yes

Reviewer #3: Yes

4. Is the manuscript presented in an intelligible fashion and written in standard English?

Reviewer #1: Yes

Reviewer #2: No

Reviewer #3: Yes

5. Review Comments to the Author

Reviewer #1: Congratulations to the authors for the excellent research. Women's experience of menopause varies hugely and there is no one-size-fits-all approach to management. Many women transition this stage of life uneventfully, whereas some experience prolonged or severe symptoms and need information, support, or medical treatment. Historically, women have been poorly served by both the research community and by society. It is time for a sensible conversation about menopause to enable informed, individualised decision making on optimal management of this transition, and several medical specialties should be part of this conversation. Understanding our population is the first step to combating misinformation, fear, and deficits in comprehending overall health and comorbidities.

Reviewer #2: Abstract

- Objective: Menopause Hormone Therapy- define the acronym MHT on first use of the abstract

Introduction:

The introduction is quite confusing, both the language and the content.

It is not clear how important it is for menopausal/premenopausal women to know their symptoms and what attitudes are recommended and how they are carried out around the world. I suggest rewriting a few paragraphs with this information that is relevant and supports your work.

We two clear objectives in our study: (1) to evaluate women’s during the menopausal transition and (2) to examine how these changes influence their decision-making regarding MHT. This second objective was included in your analysis?

“Unlike two earlier studies conducted in Brazil [18,19], before the pandemic period, our survey included premenopausal and postmenopausal patients aged between 40 and 65 years” – Did these Brazilian studies have the same objective as this study? Or did the findings of these previous studies support the decision to carry out this study?

Methods

- I suggest removing the period: “This tool was used in this study” and using the reference in the previous period.

- it is not clear that the answers to the questions are arranged on a Likert scale, nor how the authors used to assess the correlation between the answers (Reading the Results section, in Knowledge, there was a categorization to assess the degree of women’s satisfaction and the level of concerns related to menopausal period)

- the healthcare providers are only gynecologists or general practitioners?

Results

Table 1:

- organize the table, making it uniform. For example, how to put “n/N (%)” in the columns and not in the rows [Total (n (%))]

- approximate the percentages so that the total is 100%. In “Occupation” for “Premenopausal Women” the sum is 100.01%

- please, describe the statistical test used

- In the results section, it is necessary to present the findings in your research. Therefore, I suggest reviewing and removing the adverbs of intensity (surprisingly, notably).

Reviewer #3: This was a well designed study, with a expressive number of women assessed, highlighting an important theme, which will affect even more women around the world. I have only minor comments to do:

- Introduction: at the second paragraph fourth line - the utilization of this treatment use (sounds confusing as a sentence)

- Methods - also present at the abstract, the dates are not in international Standard. The year is written first, then the month, then the day. This is also known as the YYYY/MM/DD format

- At the sample size calculation there is a sentence where "864 women will be required" suggestion - 864 would be required to conduct

- Ethical considerations - Considering there are two centers, where are the approval of the second one?

Results:

At the table 1 it is written: Total %), lacking a parenthesis, and there are spaces between the parenthesis and some of the numbers - only suggestions of details

As it was done in table 2 - suggest in table 1 to put the statistical tests used

No further considerations

The discussion was brilliant, separating and organizing the writing like the way you did, only makes the reading better

6. PLOS authors have the option to publish the peer review history of their article (what does this mean? ). If published, this will include your full peer review and any attached files.

**Do you want your identity to be public for this peer review?** For information about this choice, including consent withdrawal, please see our Privacy Policy .

Reviewer #1: **Yes: ** Jan Pawel Andrade Pachnicki, Ph.D.

Reviewer #2: No

Reviewer #3: **Yes: ** Renan Massao Nakamura

---

## [Author Response · Author response to Decision Letter 1]

13 Feb 2025

Dear Editor, thank you for your email regarding our manuscript titled “Climacteric Women’s Perspectives on Menopause and Hormone Therapy: Knowledge Gaps, Fears, and the Role of Healthcare Advice.” (ID: PONE-D-24-57399). We appreciate the time and effort that you and the reviewers have dedicated to assessing our work. We are grateful for the constructive feedback provided, which will help us improve the quality and clarity of our manuscript. We worked on addressing the points raised during the review process and will submit a thoroughly revised version highlighted in yellow. I have uploaded the dataset to the stable public repository recommended by PLOS ONE: ZENODO. You can access the data by clicking the following link: https://zenodo.org/records/14867789.

Thank you once again for considering our work for publication in PLOS ONE. If there are any additional suggestions or specific guidelines we should follow during the revision process, please do not hesitate to let us know.

---

## [Editor Report · Decision Letter 1]

19 Feb 2025

PONE-D-24-57399R1Climacteric Women’s Perspectives on Menopause and Hormone Therapy: Knowledge Gaps, Fears, and the Role of Healthcare Advice.PLOS ONE

Dear Dr. Rodrigues,

Thank you for submitting your manuscript to PLOS ONE. After careful consideration, we feel that it has merit but does not fully meet PLOS ONE’s publication criteria as it currently stands. Therefore, we invite you to submit a revised version of the manuscript that addresses the points raised during the review process.

Dear Authors,

Congratulations on the corrections made to the article. The reviewers' questions were adequately answered and the changes to the manuscript are appropriate. However, I suggest correcting the following points before possible publication.

In the section “We analyzed the relationship between public and private health systems and the use of hormone therapy (HT) for menopause treatment. Our findings indicate that women using the public health system are less likely to use HT (62.59% out of 270) compared to those using the private system (49.78% out of 233), representing a difference of 12.81%. To assess the statistical significance of this observation, we performed a Chi-Square test, which yielded a *p-* value of 0.0168. Given a significance level of 0.05 (95% confidence), we can reject the null hypothesis and conclude that these results are unlikely to be due to chance. Further research will explore the broader impact of public and private health systems on treatment choices.” I believe there was a typing error, since, from what can be understood from what was written, women in the public system used more HT. I suggest reviewing it.

In Table 1, I suggest making it clear that there is missing data for some women, since the sum of the absolute numbers in the rows does not always correspond to the total number evaluated.

We look forward to receiving your revised manuscript.

Kind regards,

Luiz F. Baccaro

Academic Editor

PLOS ONE

Journal Requirements:

Additional Editor Comments:

Dear Authors,

Congratulations on the corrections made to the article. The reviewers' questions were adequately answered and the changes to the manuscript are appropriate. However, I suggest correcting the following points before possible publication.

In the section “We analyzed the relationship between public and private health systems and the use of hormone therapy (HT) for menopause treatment. Our findings indicate that women using the public health system are less likely to use HT (62.59% out of 270) compared to those using the private system (49.78% out of 233), representing a difference of 12.81%. To assess the statistical significance of this observation, we performed a Chi-Square test, which yielded a p-value of 0.0168. Given a significance level of 0.05 (95% confidence), we can reject the null hypothesis and conclude that these results are unlikely to be due to chance. Further research will explore the broader impact of public and private health systems on treatment choices.” I believe there was a typing error, since, from what can be understood from what was written, women in the public system used more HT. I suggest reviewing it.

In Table 1, I suggest making it clear that there is missing data for some women, since the sum of the absolute numbers in the rows does not always correspond to the total number evaluated.

---

## [Author Response · Author response to Decision Letter 2]

19 Feb 2025

Thank you for your thorough review and valuable feedback. We appreciate your attention to detail. We will carefully verify the data and wording to ensure accuracy and clarity.

#1. Please take a moment to review the modified text: Our results highlight an important observation regarding women’s usage of hormone therapy (HT) across different healthcare systems. Specifically, 62.59% of women who rely on the public health system (from a sample of 270) are less likely to use HT, compared to 49.78% of those in the private system (from a sample of 233) representing a difference of 12.81%. To assess the statistical significance of this observation, we performed a Chi-Square test, which yielded a p-value of 0.0168. Given a significance level of 0.05 (95% confidence), we can reject the null hypothesis and conclude that these results are unlikely to be due to chance. Further research will explore the broader impact of public and private health systems on treatment choices.

#2. Thank you for your suggestion. We acknowledge that there is missing data for some women, which is why the sum of the absolute numbers in the rows does not always match the total number evaluated. We will clarify this in Table 1 including this information as a footnote to ensure transparency.

Footnote: The absolute number of responses for each question does not represent the total sample pf 1,139 participants due to missing data.

Best Regards

Márcio Alexandre Hipólito Rodrigues

---

## [Editor Report · Decision Letter 2]

21 Feb 2025

Climacteric Women’s Perspectives on Menopause and Hormone Therapy: Knowledge Gaps, Fears, and the Role of Healthcare Advice.

PONE-D-24-57399R2

Dear Dr. Rodrigues,

We’re pleased to inform you that your manuscript has been judged scientifically suitable for publication and will be formally accepted for publication once it meets all outstanding technical requirements.

Kind regards,

Luiz F. Baccaro

Academic Editor

PLOS ONE
---

## [Editor Report · Acceptance letter]

PONE-D-24-57399R2

PLOS ONE

Dear Dr. Rodrigues,

I'm pleased to inform you that your manuscript has been deemed suitable for publication in PLOS ONE. Congratulations! Your manuscript is now being handed over to our production team.

Kind regards,

on behalf of

Dr. Luiz F. Baccaro

Academic Editor

PLOS ONE